# Tendon Stress Estimation from Strain Data of a Bridge Girder Using Machine Learning-Based Surrogate Model

**DOI:** 10.3390/s23115040

**Published:** 2023-05-24

**Authors:** Sadia Umer Khayam, Ammar Ajmal, Junyoung Park, In-Ho Kim, Jong-Woong Park

**Affiliations:** 1Department of Civil and Environmental Engineering, Urban Design and Studies, Chung-Ang University, Seoul 06974, Republic of Korea; sadiakhayam319@gmail.com (S.U.K.); pjy5451@gmail.com (J.P.); 2Department of Smart Cities, Chung-Ang University, Seoul 06974, Republic of Korea; ammarajmal@cau.ac.kr; 3Department of Civil and Engineering, Kunsan National University, Kunsan 54150, Republic of Korea

**Keywords:** tendons, prestressed girder, sensors, finite element, machine learning, neural network, dataset, artificial neural network

## Abstract

Prestressed girders reduce cracking and allow for long spans, but their construction requires complex equipment and strict quality control. Their accurate design depends on a precise knowledge of tensioning force and stresses, as well as monitoring the tendon force to prevent excessive creep. Estimating tendon stress is challenging due to limited access to prestressing tendons. This study utilizes a strain-based machine learning method to estimate real-time applied tendon stress. A dataset was generated using finite element method (FEM) analysis, varying the tendon stress in a 45 m girder. Network models were trained and tested on various tendon force scenarios, with prediction errors of less than 10%. The model with the lowest RMSE was chosen for stress prediction, accurately estimating the tendon stress, and providing real-time tensioning force adjustment. The research offers insights into optimizing girder locations and strain numbers. The results demonstrate the feasibility of using machine learning with strain data for instant tendon force estimation.

## 1. Introduction

The monitoring of precast construction is gaining attention nowadays [1]. Among precast members, prestressed structures are of the utmost importance. Prestressing is a technique in which high-strength steel tendons are stretched or tensioned to induce pre-compression in concrete. These tendons are commonly stretched between two abutments and jacked at the end [2]. Using prestressed tendons helps reduce cracking in structures that require long timespans, such as bridges, parking lots, balconies, water tanks, concrete pipes, floor slabs, and driven piles. The provision of long spans allows for uninterrupted space with fewer joints in prestressed structures. [3]. Implementing prestressing techniques in construction enhances the project’s quality by minimizing the utilization of materials and streamlining the integration of large component assembly, leading to more efficient and effective construction. The fatigue and seismic resistance of a structure can be significantly improved using prestressed concrete, as it increases the overall stiffness of the structure [4].

Prestressed concrete structures have had a significant impact on infrastructure networks. However, the prestressing process can be challenging to implement due to the need for specialized equipment, skilled labor, and strict quality control measures [5]. The lack of proper equipment and labor can also significantly affect the quality of the structure. Researchers have attempted to improve the quality of prefabrication through field tests and numerical simulations [6]. Quality control during prestressing can be achieved by carefully ensuring that the tensioning capacity and specifications meet relevant standards for use. The corrosion of steel tendons is a common cause of failure in prestressed concrete. The proper inspection of tendons can prevent this by identifying broken wires and rusting, which can reduce stress in tendons [7]. In addition, deviations in applied tensioning stress can necessitate repeating the entire prestressing process, and excessive prestressing force can cause significant cracks and blow-outs [8]. Therefore, quality control through stress monitoring during the manufacturing of precast prestressed structures is critical, as the long-term behavior of these structures can be unpredictable due to issues such as corrosion [9].

Monitoring the tendon force is essential to assessing the health and quality of prestressed concrete (PSC). Over time, the prestressing force in tendons may be reduced due to the long-term effects on prestressing steel. Variations in prestressing stress compared to design strength can result in cracks, creep, or shrinkage in the structure [10]. If this damage goes unnoticed for an extended period, it can lead to structural failure. It is particularly important to monitor tendon force in prestressed structures, as excessive creep may occur if the tendons are tensioned with a higher force than intended [11]. The behavioral effect of effective prestress was investigated in several studies [12,13]. Tanchan et al. [14] conducted a numerical analysis on unbonded tendons. They found that effective prestress has a more substantial impact on the stress increment in the tendons than on their ultimate moment capacity. These studies highlight the importance of accurately estimating stresses in tendons and their impact on various parameters.

In addition to quality control, the prestressing technique is often used to achieve the desired camber for girders by applying the required tendon force [15]. Camber is the upward deflection of a member, which tends to return to a neutral state upon applying load. The camber can be controlled by adjusting the tendon stresses [16]. Tadros et al. [17] noted that errors in estimating prestressing force and girder self-weight could lead to variability in the initial camber, causing construction challenges or the rejection of the girder. The authors suggested using modern methods for calculating the modulus of elasticity, creep, shrinkage, and other parameters in order to reduce errors in estimating the camber.

The accurate estimation of deflection and strain values is essential for evaluating the serviceability of prestressed bridges. Several methods have been used to estimate prestressing force and observe its effects on various parameters [18,19]. J. Hsiao et al. [20] studied the effect of prestressing force on midspan deflection using hand calculation. They found that neglecting the P-delta effect resulted in relatively small deflections. In contrast, by considering the moment of inertia of the gross concrete section, the calculations resulted in more considerable deflections. Ghallab et al. [21] reviewed equations for calculating the tendon stress increase at the ultimate stage, but the study was limited to externally tensioned tendons and did not examine internally stressed tendons. However, hand calculations are prone to human errors; therefore, software is preferred for fast and accurate calculations.

In general, two approaches are commonly used for monitoring prestressed structures: the direct onsite monitoring of prestressing tendons and using FEM. The onsite monitoring of prestressed girders can be carried out using strain-based methods [22,23,24,25], some of which depend on electrical impedance, vibration, and wave velocity [19]. It is challenging to apply these methods in actual construction sites and structures since the prestressed tendons are not visible from the outside and are filled with mortar. In other words, onsite monitoring necessitates direct access to prestressing tendons to be effective, limiting its applicability. The use of FEM is rapidly gaining fame and is replacing other estimation methods. Turmo et al. [26] examined the shear behavior of a prestressed concrete beam by modifying the jacking prestress using a 2D flat-joint numerical model. Other studies [27,28] examined the impact of changes in the prestressing and aspect ratios on the nonlinear structural behavior of prestressed concrete segments using FEM. Halder et al. [27] found that the stress in unbonded tendons before failure varies from 0.66 to 0.94 fpy for longer tendons using FEM, which are significantly underestimated compared to ACI318-19 code predictions [29,30]. Michael et al. [31] developed a verified 3D finite element model for a parametric study and observed the behavior of concrete bridges under different loading conditions and tendon stress increments. Other researchers have investigated corrosion, creep, and shrinkage in prestressing structures using FEM analysis [10,32]. However, the numerical method can be time-consuming and cost ineffective; consequently, more sophisticated and advanced technology is required for real-time tendon stress estimation.

A more advanced approach to estimating structural stresses in real-time is to use machine learning (ML) models, also known as surrogate models, to replace computationally expensive nonlinear numerical simulations [33]. A surrogate model that utilizes ML models is a simplified and efficient substitute for a complex and expensive model that is used in engineering and scientific simulations to make predictions and optimize designs with fewer computational resources. Creating a surrogate model is also called metamodeling in research fields other than civil engineering [34]. ML models, such as random forest regression (RFR), XGBoost, support vector machine for regression (SVR), multi-layer perceptron (MLP), and K-nearest neighbor (KNN), have been used in the past as substitutes for FEM in different fields of research [35,36,37,38]. Other approaches, such as the Kriging model, genetic algorithms [39,40], Gaussian process regression [28], and Bayesian interference framework [29], have also been used as surrogates for FE models. However, this research was limited to predicting the long-term deflections of structures, and further research is needed to explore the use of ML models for detecting tendon stresses [41]. Junkyeong et al. [42] used embedded elastic–magnetic sensors and machine learning methods to estimate tendon stresses in prestressed bridges. Giulio et al. [43] detected tendon malfunction using machine learning on monitored stress data. However, neither of these studies used a FEM-based surrogate model. Torgeir et al. [44] recently used an artificial neural network to estimate the tendon stress in unbonded members utilizing a database of experimental results from the literature. Currently, for real-time prediction, there are few or no applications of strain-based tendon stress estimation in bonded prestressed members using machine learning and FEM analysis.

The current study presents a novel framework that employs machine learning techniques to the enable immediate estimation of tendon stress based on strain data. The novelty of the research is the use of strain data from girders with bonded tendons to predict tendon stress in real time. The bridge girder’s finite FE model was created based on field observations. Three ML models, namely random forest (RF), support vector regression (SVR), and artificial neural network (ANN), were trained using the dataset obtained from the FE model, which was validated using analytical calculations. The proposed model demonstrated the most exceptional performance, and hence it was utilized to make accurate predictions of tendon stresses, affirming the effectiveness of the proposed tendon stress estimator. The ANN-based calculator enabled real-time stress estimation on new data and provided insights into the significance of various sensor permutations for optimizing the sensor number and location.

## 2. Surrogate Model Framework

The proposed tendon stress estimation model comprises two main components: FEM and machine learning (ML). The proposed framework is illustrated in Figure 1. The FEM analysis of the bridge girder was conducted by the varying tendon stress and generating data. Using the reverse engineering approach, which includes breaking down the individual components of more oversized products to extract design information, the individual strain response of the girder was used to estimate the applied stress on the structure. The FEM input was treated as the ML model output to create a surrogate model that can predict the tendon stress in real time for inputs in the form of girder strain values and an elastic modulus, as explained further in section II-D. The concept of this framework includes training the ML model on a validated FE dataset, and the trained model utilizes strain data as an input to predict tendon stresses. The input strain values were obtained for field application from strain gauges that were installed onsite at various girder locations.

In some cases, the most critical areas of a structure or material may experience high deformation or stress. It is essential that the sensors are placed in these locations to capture accurate measurements. The high-stress regions of symmetrical structures can be assumed; however, the stress distribution differs for unsymmetrical members, and the assumption cannot be made easily. Therefore, the surrogate model framework also provides information regarding the best location and number of strain gauges to be installed, based on identifying the essential features in the ML model.

### 2.1. Girder Specifications

The selection of the girder was based on two factors. Firstly, the research required using a bonded tendon girder, which was a prestressed concrete beam with bonded tendons running along its length. Secondly, the girder was selected from available girders on the field to validate the accuracy of the FEM model by comparing the results with the analytical solutions that the field experts performed.

The specific girder chosen for the study was a 45 m long I-beam girder, as illustrated in Figure 2, along with its various components. Table 1 and Table A1 in Appendix B provide detailed specifications of the girder, including its dimensions, material properties, cross-sectional area, and section modulus.

The I-beam girder featured four parabolic bonded tendons anchored at both ends. These tendons allowed the prestressed force to create a camber at the midspan of the girder. This means that the girder had a slight upward curve in the center, which helps to counteract the sag that would otherwise occur due to the weight of the girder itself and any additional loads it may be supporting.

### 2.2. FEM Modeling

The provided CAD drawings were used to develop a three-dimensional FEM model (Figure 3) of an I-beam girder using Abaqus software [45]. The software utilizes the implicit scheme to solve the problem. The model aims to simulate the actual onsite conditions and validate its accuracy by comparing the results with existing analytical solutions. The girder specifications used to develop the FEM model are provided in Table 1.

The main parts of the model consist of the concrete I-beam, steel rebars and tendons. The girder was modeled as C3D8R, an eight-node linear brick element that has six faces and four integration points per face. In terms of deformation capabilities, the C3D8R element can undergo plastic deformation, exhibit large strains, and can handle material nonlinearity. It can also model crack propagation, contact behavior, and damage evolution. Meanwhile, a T3D2, a three-dimensional, two-node truss element, was used to model the reinforcement of steel bars and pre-tensioned tendons. The interaction between the steel and concrete was modeled as surface-to-surface contact, with embedded constraint properties. The girder was modeled as a simply supported beam with one end hinged and the other end set as roller support. The diameter of the steel bars was kept at around 16 mm. The four bonded tendons were designed using a truss element following a parabolic curve and represented by embedded constraints. Prestressing was applied to each tendon using a predefined field, with the tendon stress involved in the x-direction specified as sigma11 in Abaqus FEM, along the length of the girder. A fine meshing of 50 mm in size was assigned to the structure. An implicit scheme was utilized by performing a static linear analysis under gravity load, allowing the model to deform in all directions. The details for process of girder analysis are presented in the Appendix A.

### 2.3. FEM Validation

The FEM model of the bridge girder was validated using analytical solutions obtained from the field. The formulas used to compute the tendon force ($P_{t}$), midspan stress at top fibers ($f_{ct}$), and bottom fiber ($f_{cb}$) of the I-beam girder are mentioned below:(1)$$P_{t}=f_{pt}A_{p}N$$
(2)$$f_{ct}=\frac{P_{t}}{A_{c}}-\frac{P_{t}e_{z}}{Z_{t}}+\frac{M_{d}}{Z_{t}}$$
(3)$$f_{cb}=\frac{P_{t}}{A_{c}}+\frac{P_{t}e_{z}}{Z_{b}}+\frac{M_{d}}{Z_{b}}$$

The parameters used in the above equations and their values are listed in Table 2.

The limit of initial prestressing, as per the PCI bridge design manual section 17.8.7 or ACI 318-19 article 20.3.2.5, is given as follows [30]:(4)$$f_{pt}\leq 0.8f_{pu}$$
where the $f_{pt}$ is the initial tensile stress in the tendon after prestressing or re-stressing in MPa, and $f_{pu}$ is the ultimate tensile strength of the prestressing steel in MPa. The ACI 318 code requires that the initial stress in the tendon is limited to a value that ensures that the ultimate tensile strength of the prestressing steel is not exceeded.

Based on the analytical solutions, a tendon stress of approximately 1109 MPa was applied as initial prestressing, less than the tendon stress limit of 1488 MPa provided by Equation (4). The stress in concrete immediately after the introduction of prestress to the girder and the resulting deflected shape is shown in Figure 3. The prestressing creates a camber at the midspan of the girder. This means that the girder had a slight upward curve in the center, which helps to counteract the sag that would otherwise occur due to the weight of the girder itself and any additional loads it may be supporting. The girder had a camber of approximately 35.4 mm, which falls within the limit of 50 mm, as specified by the ACI 318 standards based on the span length, specific design requirements, and loading conditions of the girder.

The analytical solutions were compared with those obtained from the finite element model, and the differences were negligible. This demonstrates the validity of the finite element model. The comparison results are summarized in Table 2. After validation, as mentioned in the following sections, the Python script from this basic FEM model served as a tool to generate data for training ML models.

### 2.4. Statistical Background of Parameters

An investigation of the statistical properties of the material strength of the concrete, rebar, and tendon strand commonly used in domestic construction sites was performed by Paik et al. [46] in order to provide a foundation of reliability-based design codes for concrete structures. The authors conducted tensile strength tests on concrete, rebar, and strand samples. They analyzed the test results using statistical methods, including the standard deviation, mean, and coefficient of variation in the material strength for each type of material. The authors found that the material strength of concrete had the highest coefficient of variation, followed by rebar and strand. The study results showed that the tensile strength of rebar and strand had a relatively low coefficient of variation, indicating that these materials have fairly consistent mechanical properties. On the other hand, a higher coefficient of variation was observed in the compressive strength of concrete, indicating that the mechanical characteristics of concrete may have more variability. Based on the literature, the parameters selected in the present research for data generation were the strain values as its strain-based stress estimation, along with concrete elastic modulus and tendon stress values.

### 2.5. Data Generation and Preprocessing

The Abaqus script was used to generate a total of 5000 data points, with the elastic modulus (EM) and tendon stress (TS) serving as input parameters, resulting in strain values at the locations S1, S2, S3, S4, S5, S6, and S7, as shown in Figure 2a. The strain locations were set based on the regions in which possible damage could occur, such as the flexural failure region at the mid and the shear failure region on supports, during overstressed conditions. Simulations were performed on a personal computer using an AMD Ryzen™ 7-5700G processor and 16 GB of operational memory.

The data generation process involved varying the tendon stress within a range of 100 to 5000 MPa and the elastic modulus within a range of 20,000 to 50,000 MPa. Using the concept of reverse engineering, the generated data were split into input and output features for the ML models. The eight input features were the strains at locations S1, S2, S3, S4, S5, S6, and S7, as well as the elastic modulus, while the tendon stress was taken as the output label.

### 2.6. Machine Learning (ML) Models

Past research has shown that the accuracy of ML models that consist of decision trees is better than those that use artificial neural networks [47]. Therefore, in the present study, the analysis was performed by increasing the number of hidden layers of an artificial neural network and using state-of-the-art ML models for comparison [48].

Supervised ML models that rely on decision tree ensembles, such as RFR, are commonly employed for regression analysis. In contrast, SVR is a supervised ML model that utilizes a hyperplane instead of a line to make predictions. In contrast, RFR does not require feature scaling [49].

This study used a grid search method to determine the optimal parameters for the ML models, including RFR and SVR. The grid search was conducted using the sci-kit-learn Python library, and the models were evaluated based on their mean square error or accuracy for each combination of hyperparameters. In addition to these models, an artificial neural network (ANN) model was also trained to compare the performance of each model in terms of percentage errors. The flow chart of the framework is shown in Figure 4, depicting the training of ML models and the selection of the model with the least MSE for the stress calculator. The stress calculator is capable of taking inputs from field sensors to estimate the tendon stress, along with providing suggestions regarding the number of strain gauges and their locations.

In this study, the ANN model’s hyperparameters were tuned to obtain a minimum mean square error score. The ANN architecture comprises a single input layer, two hidden layers, and one output layer. The number of hidden layers was determined based on the model performance and overall comparison of the model with other ML models. A total of 128 neurons for the first hidden layer and 512 neurons for the second hidden layer were finalized; refer to Figure 5 for the network architecture.

The architecture of the ANN utilizes Relu and linear activation functions. The optimization process utilized the Adam optimizer with a learning rate of 0.001, a batch size of 256, and early stopping at the lowest validation score at epoch 606. The advantage of using the ML model is its ability to make predictions very quickly compared to the FEM model, making it ideal for applications in which real-time predictions are required, such as structural health monitoring.

In order to train the ML models, the dataset was divided into training and testing subsets, where 80% of the data was allocated for training the model, and the rest was reserved for assessing its performance in the current study. The present study utilized the programming language Python and its commonly used libraries for code execution. By utilizing CPU and GPU resources, the conducted research effectively trained and evaluated the performance of the proposed ML models.

## 3. Results and Discussion

### 3.1. Prediction Comparison

In this section, the outcomes of the ML models for anticipating the tensile stress of bridge tendons are presented. The predicted values of the different ML models were in good agreement with the observed values. Figure 6 shows the prediction curves of the RFR, SVR, and ANN models for the training datasets, respectively. It can be observed that the distribution of data points for the RFR model for both the train and test datasets (Figure 6a,d) lacks a centralized regression fit line to some extent. The data points are slightly far from the diagonal line, which indicates a poor model fit. However, the model demonstrated the ability to provide reliable predictions.

The RFR model, on the other hand, showed a better fit curve of the predicted versus actual values compared to the RFR model for both the training and testing datasets, as shown in Figure 6b,e, respectively. However, the fit line is not perfectly centralized, which could potentially impact the reliability of the RFR model as a prediction tool.

In contrast, the artificial neural network model showed a better centered fit line than the other models in both the testing and training datasets (Figure 6c,f). The predictions of this model are more consistent as they are well balanced across different values.

The reliability of the ANN model for the prediction can be attributed to its ability to generalize well. In addition, a good fit model is often characterized by residual plots, where the distance of the points from the zero line is small. Thus, the normal distribution of the residual plot displayed in Figure 7a confirms that the ANN model fits the provided dataset.

Figure 7b displays the training and testing loss values as a function of epochs before early stopping. The decaying curves of the training and validation loss run parallel to each other, with a relatively small difference between them. This indicates that the model is optimal, as both the training and testing loss are minimized, and there is no evidence of over-fitting or under-fitting based on the RMSE values, as explained in Section 3.2.

Further, the performance evaluation of ML models is performed by utilizing the mean squared error (MSE) as the loss function [50], and it is calculated using Equation (5).
(5)$$MSE=\frac{1}{n}\sum_{i=1}^{n}{\left(y_{i}-{\bar{y}}_{i}\right)}^{2}$$

The coefficient of determination (*R*^2^), RMSE, and mean absolute error (MAE) are some of the popular evaluation matrices used in this study (see Equations (6)–(8)). The *R*^2^ acts as a goodness-of-fit tool by describing how well a regression line fits the data points.

Compared to other evaluation metrics, the (R) MSE increases the weight on large errors and is, therefore, more appropriate for the dataset used in this research. Additionally, the measurements obtained from RMSE are in the same units as the response variable, making it preferred for comparing the performance of ML models.
(6)$$R=\frac{\sum_{i=1}^{n}\left(y_{i}-\mu_{\varepsilon })({\bar{y}}_{i}-\mu_{p}\right)}{\sigma_{\varepsilon }\sigma_{p}}$$
(7)$$RMSE=\sqrt{\frac{1}{n}\sum_{i=1}^{n}{\left(y_{i}-{\bar{y}}_{i}\right)}^{2}}$$
(8)$$MAE=\frac{1}{n}\sum_{i=1}^{n}\left|y_{i}-{\bar{y}}_{i}\right|$$
where $y_{i}$ and ${\bar{y}}_{i}$ are experimental and predicted values; *n* is number of data points; $\sigma_{\varepsilon }$ and $\mu_{\varepsilon }$ are the standard deviation and average of the experimental values, respectively; and $\sigma_{p}$ and $\mu_{p}$ are the standard deviation and average of the predicted values, respectively.

The ML models utilized the three-fold cross-validation method to evaluate their ability to make predictions on new data, providing a more comprehensive understanding of the model’s performance.

The results of *R*^2^, RMSE, and MAE for the different ML models are summarized in Table 3. The ANN model achieved the best performance, with *R*^2^ value of 0.9984, followed by the SVR model (*R*^2^ = 0.9971) and RFR model (*R*^2^ = 0.9717). The RMSE values for the training and testing data were similar, with a slightly low RMSE value for the training data, indicating that all the ML models were optimal. None of the models in this study were under-fit or over-fit, as the models were selected after hyperparameter tuning. However, when considering the RMSE values in Table 3, the lowest error was observed for the ANN model, which also had a high *R*^2^ score. Therefore, the ANN model was selected to predict the tendon stresses in the stress calculator.

The performance of the ML models was compared with that of the finite element method model used for predicting the tensile stress of bridge tendons. The ML models were preprocessed and optimized to achieve high accuracy. All the models performed efficiently, with the ANN model showing the highest accuracy. The ANN model was then used to validate the prediction results using out-of-sample data as input values and was compared to the FEM results, which require more computation time for execution. The validation results are mentioned in Table 4. The prediction error of the models on the testing dataset was less than 10%. However, the performance of the ANN model was better for all metrics, with a percentage difference of only 0.4%, followed by SVR (2.2%) and RFR (3.02%).

### 3.2. Strain Gauge Optimization

The most important factor while observing any civil engineering structure under various loading conditions is its response in the form of strains or deflection. The strain values that show the most fluctuation in response to external loading can result in critical strain values, and the location of such critical strain values on a structure is of great importance. To observe the change in the girder strain values at different girder locations while changing the tendon stress, a graph is generated from FEM data for a particular elastic modulus value, as shown in Figure 8. It can be observed that the strain variation at the S7 strain location seems to be high, as the stress response at the midspan is more prominent compared to other positions. The pattern of variation of strain from high to low starts from S7, followed by S1, S2, S6, S3, S4 and S5. However, the variation pattern may not be the same for other input parameters; therefore, a more optimized method is required to find the preferable locations of strain gauges considering all input and output data variations.

In the present research, the inputs are mostly strain values, which are assumed to be obtained from the strain gauges attached at different locations to the bridge girder. To determine the most relevant features for evaluating the structure, the impact of each feature on the predictions of the trained model was estimated and ranked using the feature permutation importance method. This method involves randomly shuffling the values of a single feature and measuring how much the RMSE score increases as a result. Feature importance assigns a score to each feature in the provided data. The score reflects the degree of importance or relevance of the feature with respect to the output variable. The higher the score, the more significant the feature is to the output variable. As depicted in Figure 9, the feature S7 seems to have the highest RMSE score due to the reliance of the model on this feature for model prediction, followed by S5 and S2. The input features other than S7, S5 and S2 were of less importance, as shuffling the values of such features has a limited effect on the model error. The EM feature was neglected, as the motive is the optimization of the strain gauge’s location and number; therefore, only features including strain values were considered.

Based on the observations, the optimal sensor location can be predicted. For instance, the inputs S7 and S5 represent the strain values located at the mid-bottom and mid-top, respectively, of the girder (as shown in Figure 2). The middle region of a simply supported beam under load is typically prone to maximum flexural stresses, while the regions near the supports are prone to maximum shear stresses. Therefore, variations in the S7 and S5 strain values are likely to be attributed to flexural stresses, while variations in the S2 strain value (located near the supports) are likely to be attributed to shear stresses. It can be concluded that the permutation importance graph correctly identified the locations of critical strain values. The optimal number of strain gauges was decided by ranking the features based on their importance scores, and then the top features with the highest scores were selected. The value of the feature number was determined based on the desired level of feature importance and the complexity of the model. The total three number of strain gauges S7, S5, and S2 were selected as an optimal number of strain gauges. In the field, strain gauges can be attached at these critical strain positions, providing an optimal number and location of gauges for structural health monitoring purposes.

### 3.3. Strain Distribution of Girder

To ensure satisfactory performance, the stresses during the prestressing process can be customized to meet the desired level. The present research uses a range of tendon stresses to obtain the I-beam girder’s response for generating a dataset, among which the FEM contour plots are depicted in Figure 10 for a tendon stress of 1100 MPa to 3000 Mpa. The increase in applied stress increased the hogging of the girder, resulting in more tension and compression at the top and bottom fibers of the girders at the midspan, respectively. The color contour shows the high response of the girder to external forces at the anchorage and midspan location; this verifies our assumption regarding the selection of the strain gauge locations at the support and midspan. The observation of the contour plots depicts that with the aid of FEM, the prestressing forces can be customized to achieve the desired level of performance. The strains at the bottom fibers of the midspan for an applied tendon stress of 1100 MPa and 3000 MPa were −15 µε and −54 µε. The corresponding cambers were 35.6 mm and 199.7 mm since, as mentioned in section II.C, the cambers are specific for each girder based on ACI 318 [51]. Therefore, the present research makes sure that the strain and deflections are within the standards.

### 3.4. Prestress Losses

It is, however, crucial to note that the prestress force used for computing the stress will not remain constant over time, and the stresses may vary due to the increase in the concrete strength and modulus of elasticity with time. Therefore, the complete analysis and design of a prestressed concrete member must take into account the effective force of the prestressed tendon at the significant stage of loading, along with the corresponding material properties that are applicable during the relevant period of the structure’s lifespan. The most common stages in which stresses and behavior are evaluated are immediately after transferring the prestress force to the concrete section and at the service load stage, where all losses of prestress have occurred [52].

The present research deals with the first stage, that is, the initial phase of pre-tensioned prestressing in bonded tendons. For tendons and concrete, the ACI code limits the allowable stresses on both by limiting the maximum stress that the tendon exerts on the concrete. However, even if the initial stress is applied based on the limit specified in Equation (4), it is noteworthy to analyze the change in the tendon stress along the length of the girder caused by the loss resulting from the elastic shortening (ES), creep in concrete (CR), shrinkage in concrete (SH) and steel relaxation (RE) under the self-weight of the pre-tensioned prestressed girder during the initial stage, which has a duration of 30 days. The percentage loss for ES was 8%, and CR (7.5%), SH (3.2%) and RE (1.5%) were computed as per the ACI-ASCE committee 432 method [53]. The total loss (TL) was 20%, which was in accordance with the simplified computation of the total loss by PCI committee [54].

Upon cutting tendons, the initial loss of prestress in pre-tensioned concrete members occurs, and the prestressing force is transferred to the member. This transfer causes the simultaneous shortening of both the concrete and tendon, which is termed elastic shortening. The reflection of the losses during the initial stage of prestressing can be observed in the FEM analysis results, which clearly showed the reductions in the tendon stress caused by ES across the length of the girder due to the bond defined as embedded property between the prestressing steel and concrete. Figure 11 depicts the variation in the tendon stress across different sections of the prestressed I-beam girder at an initial prestressing of 1109 MPa. The tendon stress at each section is different, and the section illustrates high tendon stresses, close to what was applied, at both ends of the girder, which reduces while moving towards the midspan.

The average tendon stress at the anchorage location (support) and at the midspan were 1075 MPa and 1014 MPa; the significant loss of stress at the midspan was due to bending movements, since concrete and steel are treated as a single section once bonded, leading to a loss of 8.5%, which is close to the calculated loss of ES. Out of TL, the rest of the loss of 11.4% (127 MPa), due to other factors, was compensated by adding it to the initial prestressing and modifying the limit mentioned in Equation (4) to an Equation (9).
(9)$$f_{pt}+127MPa\leq 0.8f_{pu}$$

The reduction in prestressing across the length after the FEM analysis and after the addition of TL is depicted in Figure 12 in the form of a parabolic curve. The percentage loss from the initial stress at each section for each curve is also mentioned. The FEM results illustrate the maximum stress at the mid, and after the addition of a loss in stress, the mid tendon stress was 887 MPa, with a total loss of 20%. The observation shows that if certain strains at a particular section due to tendon stress at that section serve as input to the ML model, then after the addition of a loss in stress, the predicted tendon stress will be close to the actual initial stress being applied in the field. If the girder is monitored beyond the initial stage, where only the self-weight was considered, the difference in prediction at the initial stage and after the application of a service load can be estimated to monitor the long-term losses.

## 4. Practical Considerations

One of the promising outcomes of the developed framework is its ability to monitor the initial tendon stress so that it is within the limits, as per ACI 318-19 [30]. Ensuring limit controls the excessive creep, cracks or shrinkage, and ultimately protects the structure from failure. In addition, properly monitoring the tendon stress during prestressing provides the assurance of achieving the desired camber in the girder. Apart from advantages such as computational efficiency and cost effectiveness, the surrogate model has made it convenient to utilize strains or deflection measurements at different girder locations for the serviceability evaluation of the prestressed bridges.

To provide a more accessible method for estimating the stress in tendons during application, a more user-friendly approach is required. Therefore, for real-time tendon stress estimation, the stress estimator is proposed. The stress estimator takes, as input, the elastic modulus and strain values that are monitored in the field from the sensor and returns the tendon stress applied on the girder in real time. For safety purposes, the stress calculator considers the initial tendon stress limit provided in Equation (9). The predicted tendon stress is then compared to the stress limit to determine whether it is within the limit or not. If stress exceeds the provided limit, the user can issue a warning to check the design tendon stress during the prestressing process, ensuring quality control.

In addition, the ANN tendon stress calculator displays the sequence of strain variations from high to low for strain gauges at different locations of the girder, and an optimum number of strain gauges is suggested. The strain at S7, located at the midspan of the girder, showed the highest strain fluctuation when the tendon stress was applied. Therefore, it is convenient to place the strain gauges at locations in which the strain varies the most. The user-friendly calculator can be used prior to sensor placement to best optimize the sensor location and number.

This exploratory study presents some limitations based on simplifying a numerical model. The impact of environmental factors, such as humidity and temperature fluctuations, can influence stress and strains, which is one of the limitations of this study. The present study suggests the use of more real-world data in order to create accurate models that represent actual prestress losses.

Further, the type of girder used can differ; therefore, for each girder type, there is a requirement for an updated FE model to provide a training dataset. Once the model is trained, the tendon stress is easily estimated using the proposed framework.

The ANN-based tendon stress calculator can be modified for future use as an application that receives sensor data from the cloud, predicts the stress values, and suggests critical strain values to give a warning for safety assessment, and can propose optimal gauge number and locations.

## 5. Conclusions

The present research demonstrates the feasibility of using an ANN-based surrogate model framework to predict the stress in the tendons of a bridge girder. The research implemented a novel approach involving machine learning-based tendon stress estimation and strain gauge optimization in real time using the strain data from the FE model as inputs to a ML model.

Among other models, the ANN has the most accurately generated *R*^2^ values of 0.998 and 0.997 for the training and testing datasets, respectively. The average prediction error or MAE was less than 10% for the ANN and SVR depicting the sound prediction of the models. However, the ANN model with the lowest RMSE values demonstrated the best performance; therefore, it was selected for the surrogate model framework.

The proposed surrogate model framework using ANN estimated the tendon stress in real time, with only a 0.4% difference between it and the FEM model, where mostly strains at different girder locations were specified as inputs. The ANN-based tendon stress calculator showed a practical benefit of using the proposed framework, via the addition of the computed prestress loss (11.4%) to the predicted results. The added loss compensated for the losses expected on the field, and the estimated stress was compared with the standard limits for quality assurance and structure safety. The present framework can be utilized in situations where only strain data are available to monitor the tendon stress for the purposes of structural health monitoring.

Further, the proposed model utilizes permutation importance to select appropriate locations and the number of strain gauge to be installed. The sensor with the most strain variation was found to be located at the midspan (S7), as expected from the behavior of a simply supported girder bridge. Therefore, the sensors can be placed at locations where there is more strain variation. In the present research, three strain gauges were suggested at locations S7, S5 of the midspan, and S2 at the support, thereby providing the advantage of improved structural health monitoring and reducing unnecessary gauge installment.

The proposed surrogate model is a feasible and quick means of tendon stress estimation. The framework can be utilized for issuing warnings during the prestressing process to keep tendon stress within specified limits. In the present scenario, during the prestressing process, it is crucial to monitor the tendon stress in real time to check whether it is within the standard limits or not to avoid any consequences due to over or under-prestressing. Apart from the adaptability, the ML models are easy to use, making them accessible to a wide range of users. The research focuses on the efficient use of the ML model to replace FEM models for the tendon stress estimation of bridge girders. Tendon stress estimation benefits the industry by enhancing the safety and reliability of prestressed structures.

An application that utilizes the Internet of Things (IoT) to monitor girders during prestressing in real time is recommended as future work.

## Figures and Tables

**Figure 1 sensors-23-05040-f001:**
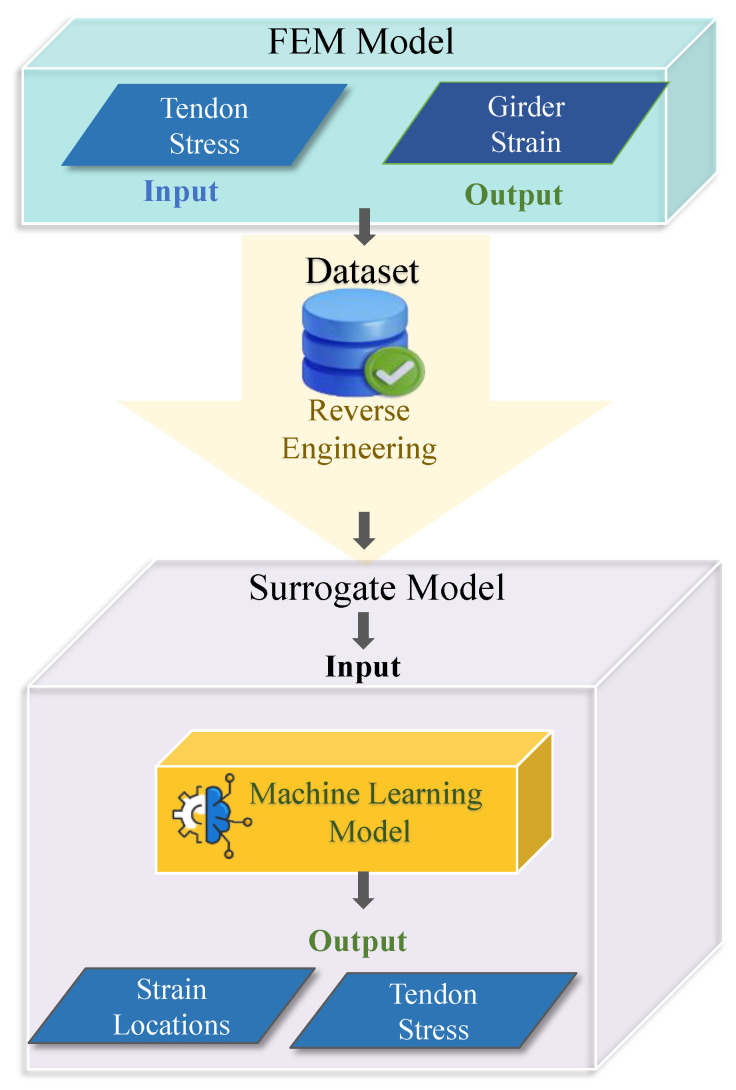
The illustration of the proposed surrogate model framework.

**Figure 2 sensors-23-05040-f002:**
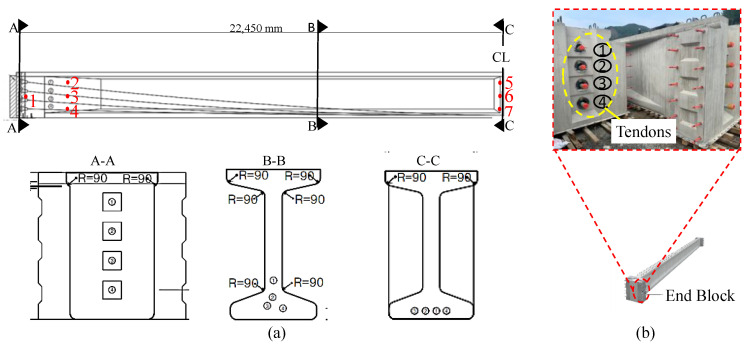
(**a**) A half-length side view of I-girder showing strain locations (1–7), along with sections at start (A-A), in-between (B-B) and middle (C-C) of girder; (**b**) Anchorage location showing four tendons at the end block.

**Figure 3 sensors-23-05040-f003:**
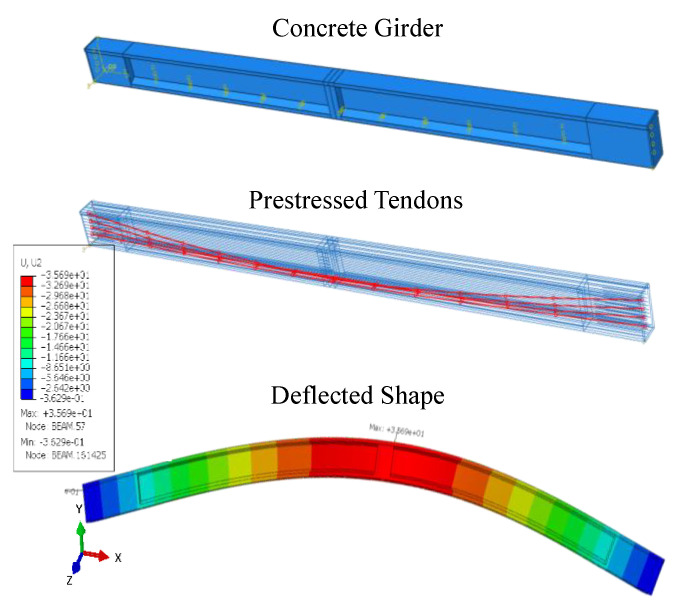
The visualization of the FEM model and analysis of the I-beam girder.

**Figure 4 sensors-23-05040-f004:**
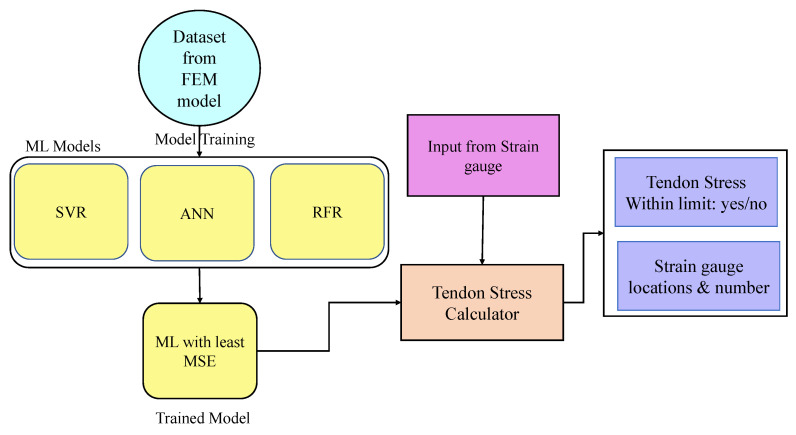
Workflow diagram of AI-based surrogate model framework.

**Figure 5 sensors-23-05040-f005:**
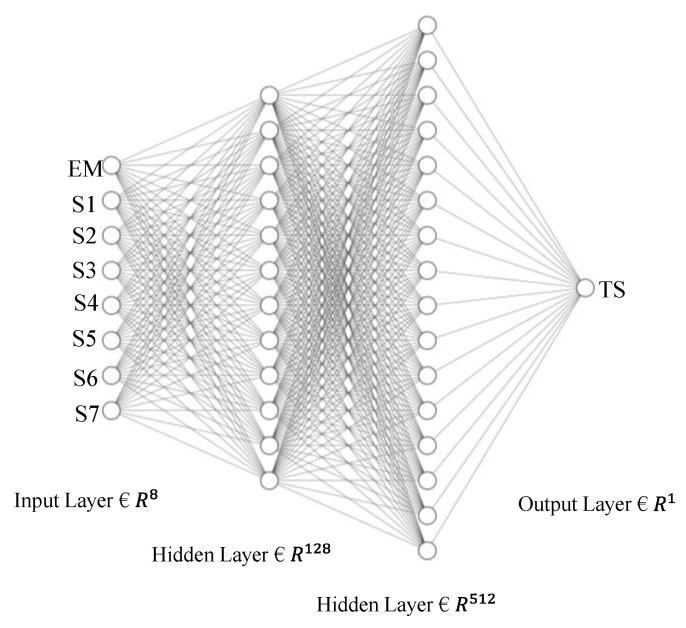
Artificial neural network architecture.

**Figure 6 sensors-23-05040-f006:**
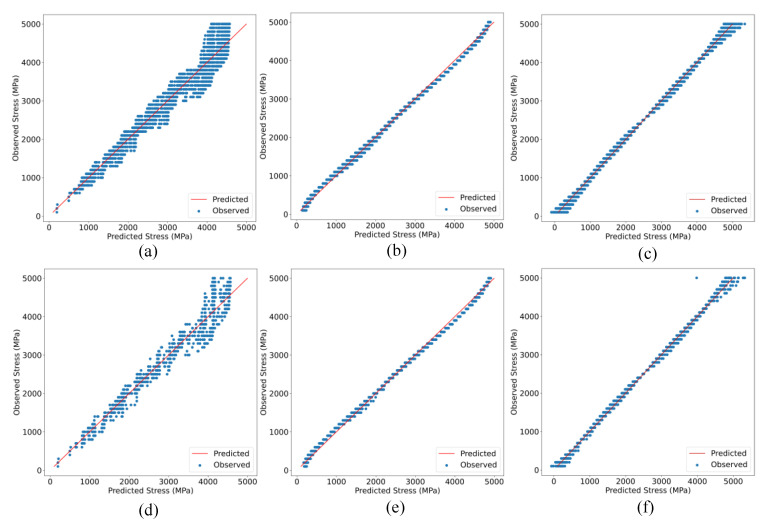
Predicted and observed values of (**a**) ANN trained correlation model, (**b**) SVR trained correlation model, (**c**) RFR trained correlation model, (**d**) ANN test correlation model, (**e**) SVR test correlation model, and (**f**) RFR test correlation model.

**Figure 7 sensors-23-05040-f007:**
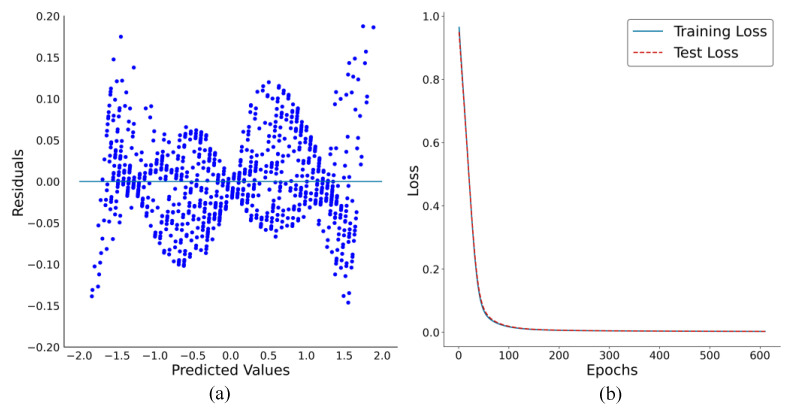
(**a**) The residual plot of ANN model, and (**b**) optimization cost of ANN model in terms of MSE.

**Figure 8 sensors-23-05040-f008:**
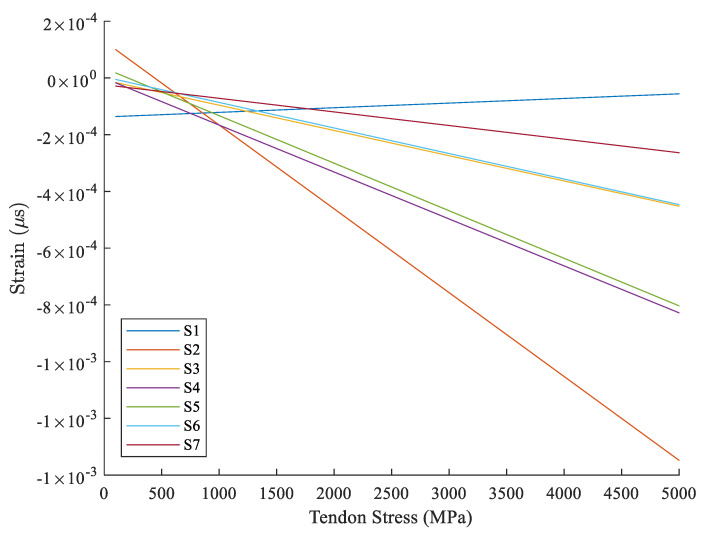
Strain variation at different locations of PCS due to applied tendon stress for an elastic modulus of 30,900 MPa.

**Figure 9 sensors-23-05040-f009:**
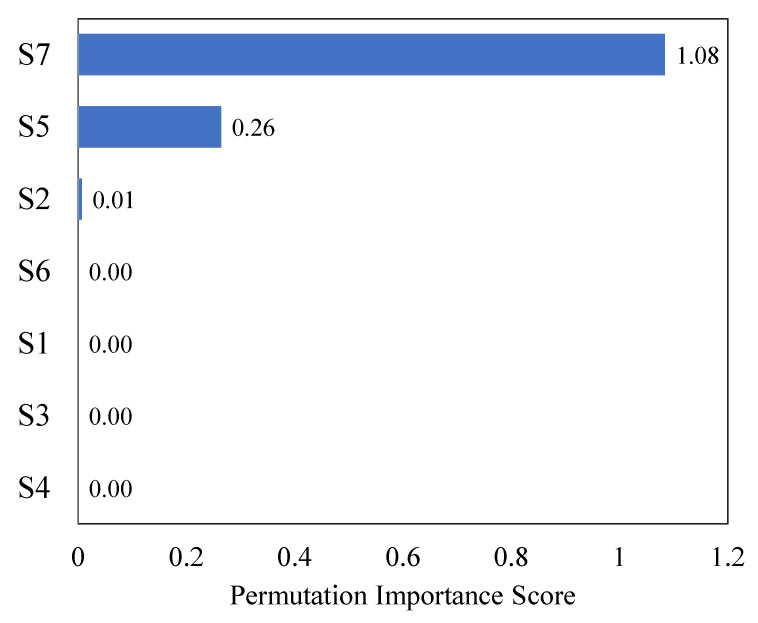
Permutation feature importance plot with increasing RMSE.

**Figure 10 sensors-23-05040-f010:**
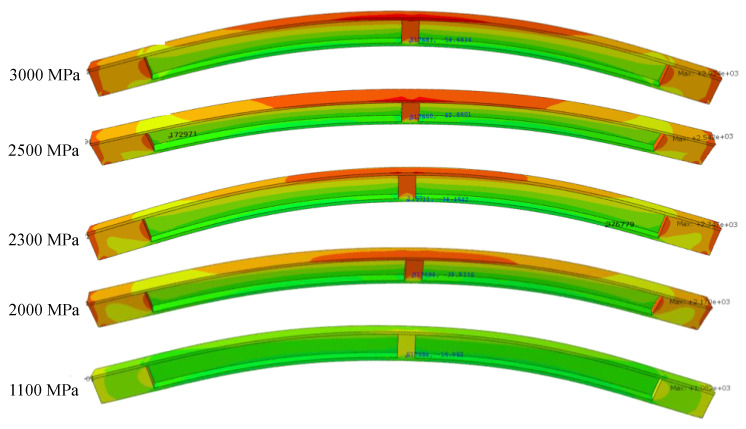
The contour plots of I-beam girder in response to a tendon stress of 1100 MPa, 2000 Mpa, 2300 Mpa, 2500 Mpa and 3000 Mpa.

**Figure 11 sensors-23-05040-f011:**
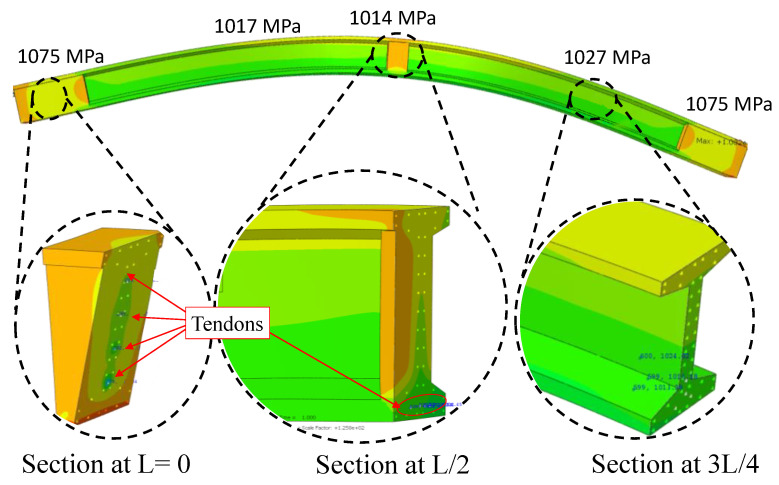
The inspection of prestressing at different sections of girder.

**Figure 12 sensors-23-05040-f012:**
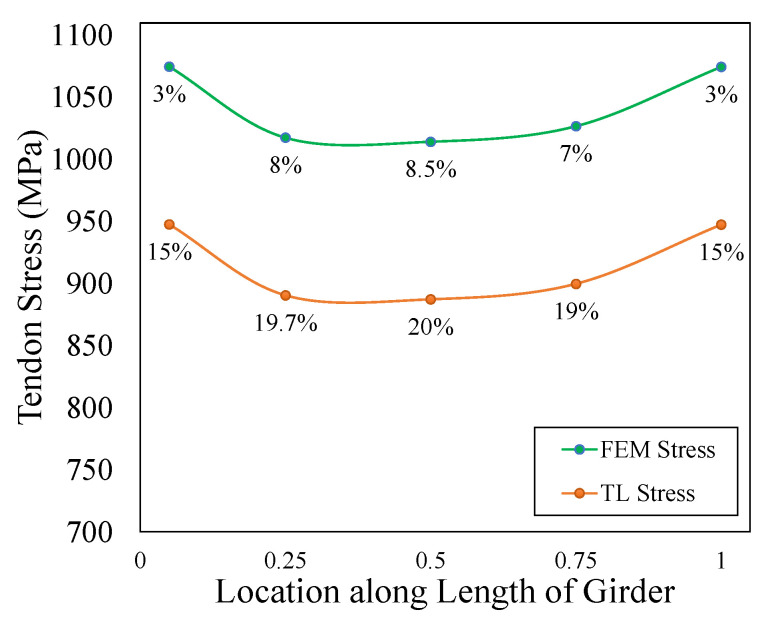
The variation in the tendon stress across the length of the girder compared to the applied prestress.

**Table 1 sensors-23-05040-t001:** Girder specifications and material properties.

Parameters	Values	Units
Steel		
Density	7.8 × 10^−6^	kg/mm^3^
Elastic Modulus	210,000	MPa
Concrete		
Density	2.1 × 10^−6^	kg/mm^3^
Elastic Modulus	30,968	MPa
Girder		
Length	45,000	mm
Height	2100	mm
Width	1300	mm

**Table 2 sensors-23-05040-t002:** A comparison of the hand calculation with the FEM model.

Beam Midspan Stresses (MPa)	Hand Calculations	FEM Model	Difference (%)
Top fiber stresses	−3.91	−3.76	3.8
Bottom fiber stresses	−17.67	−17.1	3.2

**Table 3 sensors-23-05040-t003:** Performance evaluation of ANN, RFR and SVR ML models.

	Matric	ANN	SVR	RFR
Train	R^2^	0.9984	0.9971	0.9717
	RMSE	0.0404	0.0532	0.1703
	MAE	0.0313	0.0451	0.1166
Test	R^2^	0.9978	0.9971	0.9746
	RMSE	0.0462	0.0540	0.1586
	MAE	0.0312	0.0454	0.1255

**Table 4 sensors-23-05040-t004:** The comparison of ML model prediction with FEM model for validation.

Model	Tendon Stress	Difference (%)
ANN	1305	0.4
SVR	1272	2.2
RFR	1339	3.0
FEM	1300	

## Data Availability

The data presented in this study are available on request from the corresponding author. The data are not publicly available due to agreements with the funding agency.

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
