# Peer review of "Tendon Stress Estimation from Strain Data of a Bridge Girder Using Machine Learning-Based Surrogate Model"

_sensors, 2023, doi:10.3390/s23115040_

Round 1

Reviewer 1 Report

In this paper ANN was used to develop a prediction model to evaluate the tendon strain in the prestressed beams. The paper is good and suitable for publication after considering the following comments:

·        Please add some information about the limitations of current literature, why we need this model and what is the new and novelty of this research.

·        Please also talk about why this work is important, the strain in tendon really can be evaluated using different methos such as analytical and design equations, finite element models and experimental testing. The first two are handy and easy to do. Why we this prediction model.

·        Please give more details about the finite element modes such as material models, nonlinearities, mesh size and solution mechanism. You see similar descriptions in these papers “Parametric Study on Steel–Concrete Composite Beams Strengthened with Post-Tensioned CFRP Tendons”, “Effect of external post-tensioning on steel-concrete composite beams with partial connection” and “Pre-damage effect on the residual behavior of externally post-tensioned fatigued steel-concrete composite beams”.

·        In the conclusion part please write something about how this work will be beneficial for the industry.

Author Response

The authors deeply appreciate the reviewers for giving us valuable comments that are helpful to improve quality of the paper. The authors have tried to address all the comments

Reviewer #1

In this paper ANN was used to develop a prediction model to evaluate the tendon strain in the prestressed beams. The paper is good and suitable for publication after considering the following comments:

Comments:

  1. Please add some information about the limitations of current literature, why we need this model and what is the new and novelty of this research.

Answer: The authors would like to thank reviewer# 1 for pointing this out. The comment has been addressed by adding limitations of current literature and novelty of present research in the introduction part, line 119-125 and the conclusion part, line 594-597

  1. Please also talk about why this work is important, the strain in tendon really can be evaluated using different methods such as analytical and design equations, finite element models and experimental testing. The first two are handy and easy to do. Why we this prediction model.

Answer: The authors acknowledge the reviewer’s comment. These methods are usually time-consuming as mentioned in the literature part lines 82-85, 102-104 Therefore, for fast and real-time tendon stress estimation, a machine learning model is trained. Further, the advantages of the present work is also mentioned in lines 636-643.    

  1. Please give more details about the finite element modes such as material models, nonlinearities, mesh size and solution mechanism. You see similar descriptions in these papers “Parametric Study on Steel–Concrete Composite Beams Strengthened with Post-Tensioned CFRP Tendons”, “Effect of external post-tensioning on steel-concrete composite beams with partial connection” and “Pre-damage effect on the residual behavior of externally post-tensioned fatigued steel-concrete composite beams”.

Answer: The authors appreciate reviewer’s comment on the FE details and provided suggestions. The comment is addressed by adding more details of FE analysis and properties in line 185-203.

  1. In the conclusion part please write something about how this work will be beneficial for the industry.

Answer: The authors would like to thank reviewer# 1 for pointing this out. The comment is addressed by adding the beneficial aspect of research in the conclusion. (lines 636-643) 

Reviewer 2 Report

In the Reviewer opinion the research paper entitled “Tendon Stress Estimation from strain data of a Bridge Girder using Machine Learning-Based Surrogate Model” is very good.

This paper utilizes a strain-based machine learning method to estimate real-time applied tendon stress. A dataset was generated using finite element method (FEM) analysis, varying the tendon stress in a 45-meter girder. Network models were trained and tested on various tendon force scenarios, with prediction errors less than 10%. The model with the lowest RMSE was chosen for stress prediction, accurately estimating tendon stress and providing real-time tensioning force adjustment. The re- search offers insights into optimizing girder locations and strain numbers. The results demonstrate the feasibility of using machine learning with strain data for instant tendon force estimation.

Some comments which greatly enhance the understanding of the paper and its value are presented below. Specific issues that require further consideration are:

  1. The title of the manuscript is matched to its content.
  2. The Introduction generally covers the cases.
  3. The methodology was clearly presented.
  4. In the Reviewer’s opinion, the current state of knowledge relating to the manuscript topic has been presented.
  5. Experimental program and results looks interesting and was clearly presented.
  6. In the Reviewer’s opinion, the bibliography, comprising 55 references, is representative.
  7. An analysis of the manuscript content and the References shows that the manuscript under review constitutes a summary of the Author(s) achievements in the field.
  8. In the Reviewer’s opinion the manuscript is well written, and it should be published in the journal.

no comments

Author Response

Reviewer #2

In the Reviewer opinion the research paper entitled “Tendon Stress Estimation from strain data of a Bridge Girder using Machine Learning-Based Surrogate Model” is very good.

This paper utilizes a strain-based machine learning method to estimate real-time applied tendon stress. A dataset was generated using finite element method (FEM) analysis, varying the tendon stress in a 45-meter girder. Network models were trained and tested on various tendon force scenarios, with prediction errors less than 10%. The model with the lowest RMSE was chosen for stress prediction, accurately estimating tendon stress and providing real-time tensioning force adjustment. The re- search offers insights into optimizing girder locations and strain numbers. The results demonstrate the feasibility of using machine learning with strain data for instant tendon force estimation.

Some comments which greatly enhance the understanding of the paper and its value are presented below. Specific issues that require further consideration are:

Comments:

  1. The title of the manuscript is matched to its content.
  2. The Introduction generally covers the cases.
  3. The methodology was clearly presented.
  4. In the Reviewer’s opinion, the current state of knowledge relating to the manuscript topic has been presented.
  5. Experimental program and results looks interesting and was clearly presented.
  6. In the Reviewer’s opinion, the bibliography, comprising 55 references, is representative.
  7. An analysis of the manuscript content and the References shows that the manuscript under review constitutes a summary of the Author(s) achievements in the field.

In the Reviewer’s opinion the manuscript is well written, and it should be published in the journal.

Answer: The authors are thankful for all the appreciation from reviewers. The manuscript is further enhanced by addressing all the comments from reviewers.

Reviewer 3 Report

Notes in the attachment.

Author Response

Reviewer #3

Comments:

  1. Present the essence of the surrogate model, its basic properties and advantages.

Answer: The authors acknowledge reviewers’ comment, the details of surrogate model are present in line 108-111.   

  1. Figure 3, enlarge the legend.

Answer: The comment is addressed and the legend in figure 3 is enhanced.

  1. Where does the value of 5000 tendon stress comes from while giving range of stress variation from 100 to 5000 MPa, give justification from engineering point of view. See also fig.6 and 9 in comparison with fig 8.

Answer: The 5000 MPa is the final range of tendon stress to generate a dataset of 5000 points; the selection of this range was random to generate more variations in datapoints for better prediction. However, the limit assigned to the final prediction will always give results within ultimate tendon stress. The figure 6 and 9 depict the results of data generated from FEM with the given range, but figure 8 shows the validation results that are within limits.          

  1. chapters:- 2.6, 2.7, 2.8-3.1,2, 3.3 and 3.4 combine and shorten to the minimum necessary (up to 1, 2 paragraphs) 2- too basic information - they contribute little to the article

Answer: The comment is addressed by briefing and combining 2.6 to 2.8 sections as one section and combining 3.1 to 3.3 sections.  

  1. I suggest replacing the figure 8 with a table.

Answer: The comment is addressed by adding a table 4 instead of figure 8.

  1. Describe the figure 13.

Answer: The comment is addressed and the description of figure 13 is highlighted in lines 549-557.

  1. Chapter 4 and Figure 14 are of little use and do not add knowledge to the article

- delete as chapter

- replace with a paragraph

Answer: The authors agree with your point; therefore, the suggestion is incorporated in the revised script by incorporating the paragraph instead of the chapter lines 598-613.

  1. remove the attachment (Appendix A)

- little useful data

- if necessary, integrate it into the text of the article in a less developed form

Answer: The reviewer’s comment is addressed by removing table A1 in Appendix A.(lines 691-698)